# Immunity to Sda1 Protects against Infection by Sda1^+^ and Sda1^−^ Serotypes of Group A *Streptococcus*

**DOI:** 10.3390/vaccines10010102

**Published:** 2022-01-11

**Authors:** Shuai Bi, Jie Wang, Meiyi Xu, Ning Li, Beinan Wang

**Affiliations:** 1Key Laboratory of Pathogenic Microbiology and Immunology, Institute of Microbiology, Chinese Academy of Sciences, Beijing 100101, China; bish@im.ac.cn (S.B.); wangjie@im.ac.cn (J.W.); xumy@im.ac.cn (M.X.); lining@im.ac.cn (N.L.); 2Savaid Medical School, University of Chinese Academy of Sciences, Beijing 100049, China

**Keywords:** group A *streptococcus*, DNases, Sda1, neutrophil extracellular traps, pharyngeal colonization

## Abstract

Group A *Streptococcus* (GAS) causes a variety of diseases globally. The DNases in GAS promote GAS evasion of neutrophil killing by degrading neutrophil extracellular traps (NETs). Sda1 is a prophage-encoded DNase associated with virulent GAS strains. However, protective immunity against Sda1 has not been determined. In this study, we explored the potential of Sda1 as a vaccine candidate. Sda1 was used as a vaccine to immunize mice intranasally. The effect of anti-Sda1 IgG in neutralizing degradation of NETs was determined and the protective role of Sda1 was investigated with intranasal and systemic challenge models. Antigen-specific antibodies were induced in the sera and pharyngeal mucosal site after Sda1 immunization. The anti-Sda1 IgG efficiently prevented degradation of NETs by supernatant samples from different GAS serotypes with or without Sda1. Sda1 immunization promoted clearance of GAS from the nasopharynx independent of GAS serotypes but did not reduce lethality after systemic GAS challenge. Anti-Sda1 antibody can neutralize degradation of NETs by Sda1 and other phage-encoded DNases and decrease GAS colonization at the nasopharynx across serotypes. These results indicate that Sda1 can be a potential vaccine candidate for reduction in GAS reservoir and GAS tonsillitis-associated diseases.

## 1. Introduction

Group A *Streptococcus* (GAS) species cause more than 700 million human infections each year [1], resulting in over half a million deaths globally [2]. The development of a commercial GAS vaccine is hampered by the presence of multiple GAS serotypes and infection-associated autoimmune diseases [3]. GAS is the most common bacterial cause of pharyngitis, with over 600 million cases per year [4]. GAS pharyngitis appears to be the primary reservoir responsible for the maintenance and transmission of GAS to a new host [2]. Thus, a safe and effective human vaccine holds the promise of reducing disease burden.

Neutrophils act as the first line of defense against microorganisms. In response to inflammatory stimuli, neutrophils migrate from the circulating blood to infected tissues, where they efficiently eradicate pathogens through phagocytosis, release of reactive oxygen species, and degranulation [5]. Aside from these well-documented mechanisms, neutrophils are also able to extrude neutrophil extracellular traps (NETs), which consist of DNA lattices composed of antimicrobial molecules, to entrap and facilitate the killing of microbial pathogens [6]. Deficiency in NET release or dismantling the NET backbone by bacterial DNases renders the host susceptible to infections [7]. The DNases of GAS have been identified as virulence factors and contribute significantly to disease progression by protecting GAS against neutrophil killing through degrading of NETs [8]. A total of eight GAS DNase genes have been identified to date. Six DNase genes are known to be associated with integrated prophages. Two are chromosomally encoded, and one of these is cell-wall anchored. There is homology between DNases of different GAS strains and also streptococcal species [9]. At least one of the DNases is produced by all strains of GAS and multiple DNases can be produced by a single GAS strain [8].

Sda1 is a phage-encoded DNase and shares high homology with other secreted DNases in GAS [10]. Sda1 promotes bacterial survival in neutrophil- and blood-killing assays and enhances virulence in a mouse model of invasive GAS infection. DNase inhibitors increase GAS sensitivity to neutrophil killing [11], whereas over-expression of Sda1 facilitates bacterial dissemination in vivo [12]. In addition, Sda1 suppresses the Toll-like receptor 9-mediated innate immune response through the degradation of bacterial DNA [13] and impairs plasmacytoid dendritic cell recruitment by reducing type I interferon levels at the site of infection [14]. Further studies found that Sda1 is linked to enhanced virulence of the globally disseminated M1T1 clone [12,15,16]. These results indicate that Sda1 contributes significantly to GAS virulence and pathogenesis.

Pooled intravenous immunoglobulin (IVIG) with specific anti-Sda1 antibodies have been reported to abolish Sda1 activities [17], suggesting that inhibition of DNase activity of Sda1 contributes to control of GAS infection. The nasal-pharyngeal mucosa is a common site of GAS colonization. Mucosal immunization can elicit an immune response at mucosal sites besides the serum and efficiently reduces GAS colonization in the pharyngeal cavity [18,19]. Herein, we report that intranasal (i.n.) immunization of mice with Sda1 induced antibody responses that effectively prevented degradation of neutrophil-formed NETs and pharyngeal colonization by Sda1^+^ and Sda1^-^ GAS strains with different serotypes.

## 2. Materials and Methods

### 2.1. Ethics Statement

This study was performed in strict accordance with the recommendations in the Guide for the Care and Use of Laboratory Animals of the IMCAS (Institute of Microbiology, Chinese Academy of Sciences, Bejing, China) Ethics Committee. The protocols were approved by the Committee on the Ethics of Animal Experiments of IMCAS (permit number: APIMCA2019004). All animal experiments were conducted under isoflurane anesthesia, and all efforts were made to minimize suffering of animals employed in this study. The collection and use of human serum samples were approved by the Ethics Committee of Beijing Children’s Hospital. Informed consent was obtained from the human subjects prior to enrollment in the study.

### 2.2. Bacterial Strains and Culture Conditions

GAS serotype M1 (strain 90-226) was obtained from the University of Minnesota. Serotypes M3.11, M8, M12, M28, and M49 were clinical isolates obtained from Beijing Children’s Hospital. All strains were grown in Todd-Hewitt broth supplemented with 2% neopeptone (THB-Neo; BD Bioscience, San Jose, CA, USA) at 37 °C in 5% CO_2_. Overnight cultures were pelleted, washed, and resuspended in PBS and then used for mouse challenge studies. CFUs were verified by plating on blood agar plates.

### 2.3. Polymerase Chain Reaction Amplification for the Detection of the Sda1 Gene

Genomic DNA was extracted from the GAS isolates using TGuide Bacteria Genomic DNA Kit (Tiangen, Beijing, China) according to the manufacturer’s instructions. The extracted genomic DNA was used as template, and the following set of primers targeting *sda1* were used to amplify a 684 bp fragment: *sda1*-forward (5′-TTCCCGAACTTTATCGTACAA-3′) and *sda1*-reverse (5′-CAGTAGAAGATAAGAGTCCACCG-3′) [20]. The *SrtA* gene (GenBank accession no. SQG52106.1) was amplified as a positive control with the following set of primers to amplify a 495bp fragment: *SrtA*-forward (5′-GTCTTGCAAGCACAAATGGC-3′) and *SrtA*-reverse (5′-CTAGGTAGATACTTGGTTA-3′) [21]. The products were run on a 1% agarose gel (*w*/*vol*) for visualization.

### 2.4. Cloning of Recombinant Protein

The cDNA encoded Sda1 was amplified by PCR from GAS M1 (strain 90–226). Wt-Sda1 (amino residues glutamic acid 38 to glutamic acid 390) was cloned into the Nco I and Xho I restriction sites of pET28a (+) vector. Enzymatically inactive Sda1 mutant (Mu-Sda1) was generated with an amino acid site mutation (histidine 188 replaced with glycine) [16]. The cDNA encoded DNase B (amino residues Arginine 43 to Lysine 271, NCBI Reference Sequence: WP_010922721.1) was amplified by PCR from the GAS M3.11 genome. The amplified PCR product was cloned into the Nde I and Xho I restriction sites of pET28a (+) vector. The resulting constructs were transformed into *Escherichia coli* BL21(DE3) for expression. Recombinant proteins were expressed and purified as previously described [22]. Recombinant DNase B was treated with thrombin protease at a concentration of 2 units/mL for 12 h at 20 °C to remove the N-terminal His-tag. Bacterial lipopolysaccharide was removed following purification (≤0.1 EU/μg) [19].

### 2.5. Western Blotting

Forty milliliters of overnight bacteria culture supernatants were obtained by centrifugation and filtration through a 0.22 μm filter. Proteins were precipitated through the addition of 3 volumes of ice-cold ethanol with the protein pellet resuspended in 500 μL of PBS. Equal loading was confirmed by measurement of total protein with the BCA protein Kit (Tiangen Biotech, Beijing, China). Proteins were separated by a 12% Tris·HCl SDS/PAGE gel (Bio-Rad, Hercules, CA, USA) and then were transferred to a PVDF membrane (Merck Millipore, Darmstadt, Germany). Membranes were blocked with 1% BSA (Sigma Aldrich Inc., St. Louis, MO, USA) before incubation with Mu-Sda1 immunized mouse sera (3 × 10^3^-fold dilution). Bound antibody was detected by using goat anti-mouse secondary antibody conjugated to horseradish peroxidase (HRP)-conjugate (Southern Biotech, Birmingham, AL, USA). The antibody-recognized band was visualized by ECL Western Blotting Substrate (Pierce, Thermo Fisher Scientific, Waltham, MA, USA). For detection of cross-reaction of anti-Sda1 with recombinant DNase B and Homo sapien DNase I (Sino Biological, Beijing, China), the protein samples were separated by a 12% Tris·HCl SDS/PAGE gel, Western blotting assays were performed with mouse anti-Mu-Sda1 serum.

### 2.6. Mouse Immunization and Challenge Experiment

Female C57/B6N mice (aged 4 to 6 weeks) were purchased from Vital River Laboratory Animal Technology (Beijing, China) and maintained under specific-pathogen-free conditions. Mice were anesthetized with an isoflurane/oxygen mixture and i.n. inoculated with 10 μg of Mu-Sda1 combined with 10 μg of CpG (CpG-oligodeoxynucleotides 1826/sequence 5′-TCCATGACGTTCCTGACGT-3′ (Generay Biotechnology, Shanghai, China) as an adjuvant, in a 10 μL volume/mouse (5 μL per nostril). Control mice were administered 10 μg of CpG in phosphate buffered saline (PBS). Mice i.n. infected with a low dose (approximately 0.5–1 × 10^8^/mouse) of live GAS M1 strain were used in the infection control group. Mice were immunized or infected three times at 1-week intervals. In the mucosal challenge model, mice were i.n. challenged with GAS M1 strain or serotype M12 or M3.11 at a sublethal dose of 2.0 × 10^8^/mouse. The mice were euthanized, and nasal-associated lymphoid tissues (NALT) were taken 2 weeks after the last immunization. The NLAT were homogenized in PBS and plated on blood agar plates after serial dilution to determine colony forming units (CFUs) [23]. For the systemic infection model, mice were challenged i.n. with a lethal dose (3 × 10^8^/mouse) of the GAS M1 strain, and weight loss and survival were monitored every day for 14 days. Mice with weight loss of 25% of the starting body weight were euthanized and registered as dead. Mortality was an anticipated outcome and was approved by the animal ethics committee. Surviving mice were humanely euthanized at the end of the experiment.

### 2.7. Enzyme-Linked Immunosorbent Assay (ELISA) for Antibodies

Antigen-specific antibodies were measured by ELISA as previously described [19]. Plates were coated with 5 μg/mL of single recombinant antigen in PBS and incubated overnight at 4 °C. Aliquots (100 μL) of each serum sample or mouth wash were added to each well of plates and incubated at 37 °C for 2 h. Horseradish peroxidase (HRP)-conjugated goat anti-mouse IgG or goat anti-mouse IgA were used to detect mouse antibodies. HRP-conjugated goat anti-human IgG was used for the detection of human serum IgG. A standard curve was generated by adding double diluted purified mouse IgG (Alpha Diagnostic Int, Paramus, NJ, USA) or IgA (Bethyl Laboratories, Waltham, MA, USA) to anti-mouse IgG or anti-mouse IgA coated wells. ELx800 plate reader (BioTek, Santa Clara, CA, USA) was used to measure absorbance at 450 nm, and the absorbance at 630 nm was used as the internal control. The concentration of the antibody levels was calculated according to the standard curve.

### 2.8. Preparation of Rabbit Serum IgG against Sda1

Female New Zealand white rabbits (specific pathogen-free, 2.0–2.5 kg) were purchased from JinMuYang Laboratory Animal Center (Beijing, China) and kept in the SPF animal facility for 1 week. Rabbits were subcutaneously immunized with recombinant Mu-Sda1 (5 mg) mixed with 300 μL incomplete Freund’s adjuvant (Becton Dickinson, Franklin Lakes, NJ, USA). Control rabbits were administered with incomplete Freund’s adjuvant in an equal volume. Rabbits were immunized three times at 2-week intervals. Blood samples were harvested two weeks after the last immunization. Serum IgG was purified using Protein A MagBeads (GenScript, Nanjing, China) for NETs degradation assay.

### 2.9. Plasmid DNA Degradation Assay

Overnight GAS culture supernatants were filtered through a 0.22-μm-pore-size filter. Plasmid DNA (2.0 μg) was combined with recombinant Wt-Sda1(2 μg in 3 μL), 1 × 10^3^-fold dilution of GAS culture supernatant (3 μL), or pre-incubated with an equal volume of mouse immunized serum to a final volume of 30 μL reaction buffer (20 mM Tris, 0.1 mM MgCl_2_) for 15 min at 37 °C. To halt DNase activity, 0.33 M EDTA (12.5 μL) was added to the reaction. Relative DNA degradation was visualized using a 1% agarose gel.

### 2.10. NETs Degradation Assay

Mouse neutrophils were isolated from the peripheral blood using Mouse Peripheral Blood Neutrophil Separator Kit (TBD, Tianjin, China) according to the manufacturer’s instructions. The isolated neutrophils were seeded at 2 × 10^5^ in poly-L-lysine pre-coated glass bottom dishes in RPMI 1640 supplemented with 2% fetal calf serum and incubated for 30 min at 37 °C and 5% CO_2_. The neutrophils were stimulated with 100 nM phorbol myristate acetate (PMA) for 4 h to induce NET formation. GAS culture supernatant (10^3^-fold dilution) or GAS pre-incubated with serum IgG from Mu-Sda1 immunized rabbit. Serum IgG from adjuvant immunized rabbit was used as naïve control. Wells containing only medium were used to confirm the formation of NETs. The neutrophils were incubated for a further 15 min at 37 °C and 5% CO_2_. A solution of 0.33 M EDTA was added to the reaction to stop DNase activity. After washing with PBS, the neutrophils were fixed in 4% paraformaldehyde/PBS for 30 min and washed again with PBS. DAPI (Thermo Fisher Scientific, Waltham, MA, USA) was added for a 10-min incubation and fluorescence was measured using Laser Scanning Spectral Confocal Microscope (Leica Microsystems GmbH, Mannheim, Germany).

### 2.11. Statistical Analysis

Statistical analyses were performed using GraphPad Prism software, version 7.04 (GraphPad Software, San Diego, CA, USA). The CFUs were analyzed by two-tailed unpaired Mann–Whitney *U* nonparametric *t*-tests. In addition, other variables were compared by two-tailed Student’s *t* test. One-way ANOVA with Tukey’s post-test was used to analyze the statistical significance among more than two groups. Survival curve was analyzed by the Log-rank test. A *p*-value of <0.05 was considered statistically significant. Data were presented as means ±SEM.

## 3. Results

### 3.1. Sda1 Prevalence among GAS Isolates and Antibody Response to Sda1 in Mice and Humans

Invasive GAS disease is associated with acquired strains carrying prophages encoding DNase Sda1 [8,24]. The Sda1 gene was reported in 135 isolates of 141 emm12 GAS clinical strains from Hong Kong clinical isolates [25]. Blast search result indicates that the identity of Sda1 amino acid sequence is >77.64% with other strains, including GAS strains JRS4, RS4, MGAS8232, UTMEM-1, ATCC 10782, ATCC 10782, ATCC 10782, ATCC 10782, UTSW-2, UTSW-2, and MGAS2096. This finding indicates that Sda1 commonly presents in these bacteria. PCR assays were conducted to determine Sda1 prevalence in GAS among the clinical isolates collected in the laboratory. The results showed that *SrtA*, a conserved gene ubiquitously present in GAS, was detected in all six tested strains, and the *Sda1* gene was found in five of these strains (serotypes M1, M8, M12, M28, and M49) but not in serotype M3.11 (Figure 1A). The same results at the Sda1 protein level were confirmed by Western blotting analysis (Figure 1B), indicating that Sda1 is commonly expressed in clinical GAS isolates. Recombinant Sda1 (Wt-Sda1) and enzymatically inactive Sda1 mutant (Mu-Sda1) were generated (Materials and Methods) [13,16] and their DNase activity was determined by plasmid DNA degradation assay. Wt-Sda1, but not Mu-Sda1, degraded plasmid DNA in a time-dependent manner (Figure 1C), indicating that the DNase activity of Mu-Sda1 was abolished.

### 3.2. Antibody Response to Sda1 in Mice and Humans

To estimate the potential utility of the Mu-Sda1 as a vaccine candidate, mice were immunized with Mu-Sda1 i.n. and samples of serum and mouthwash from the mice were taken to determine the immunogenicity of Mu-Sda1. ELISA revealed that antigen-specific serum IgG and IgA, and IgA in the mouthwash samples were significantly increased in the immunized mice (Figure 2A–C). The antibody also recognized Wt-Sda1 (Figure 1). However, anti-Sda1 antibody was not detected in mice that were infected with GAS (Figure 2A–C). To determine the antigenicity of Sda1 in humans, serum samples from 49 children (aged 5–15 years) were examined. The human sera highly responded to heat-killed GAS (HK-GAS) and streptolysin O (SLO), a toxin secreted by nearly all clinical GAS isolates tested [26]. However, the antibody response to Sda1 was as low as the response to keyhole limpet hemocyanin (KLH), a non-microbial related protein (Figure 2D). Nondetectable Sda1 antibody in GAS-infected mice and the limited anti-Sda1 response in humans indicated that the humoral response to Sda1 is restrained in natural GAS infection.

### 3.3. Anti-Sda1 Serum Neutralized Sda1-Inhibited Neutrophil NETosis

A collection of serum samples from immunized mice were tested for their ability to neutralize the DNase activity of Sda1 by plasmid degradation assay. A clinical isolate of GAS (M1) was grown in vitro to the stationary phase, and its culture supernatant was pre-incubated with or without anti-Sda1 sera. Next, plasmid DNA was added in the culture supernatant. The supernatant or Wt-Sda1 completely degraded plasmid DNA when incubated with PBS (Figure 3A, lanes 3 and 4) or with serum from naïve mice (Figure 3A, lanes 5 and 6). The digestion was prevented when the culture supernatant or Wt-Sda1 was pre-incubated with mouse anti-Sda1 serum (Figure 3A, lanes 7 and 8). The serum from GAS-infected mice did not rescue the digestion (Figure 3A, lanes 9 and 10), which is consistent with the lack of Sda1-specific antibody in these mice. Sda1 is a virulence factor that protects GAS against neutrophil killing by degrading NETs [11,13]. The NETs degradation assay with mouse neutrophils revealed that NETs appeared following a 4-h stimulation with PMA and disappeared following the addition of Wt-Sda1. NETs were still present after adding rabbit anti-Sda1 serum IgG-pretreated Wt-Sda1 (Figure 3B), indicating neutralizing activity of the antiserum.

### 3.4. Immunization with Sda1 Prevented GAS Colonization in the Nasopharynx

Pharyngitis is the most common GAS infection and NET degradation by GAS DNases may contribute to the establishment of pharyngeal infection. The efficient rescue of Sda1-prevented NETosis by anti-Sda1 suggests that Sda1-induced immune response would provide protective immunity against GAS infection. Mice were grouped and administrated i.n. with Mu-Sda1 with CpG or CpG (adjuvant control) or infected with live GAS three times at weekly intervals. Two weeks after the last immunization, all groups were i.n. challenged with GAS (M1). The number of CFUs in NALT was determined 24 h after the challenge. CFUs were substantially lower in Sda1-immunized mice, compared with adjuvant control mice, and were comparable to the mice that recovered from previous infection (Figure 4A). The prevalence of Sda1 in GAS clinical isolates suggested that anti-Sda1 immunity would provide protection across different GAS serotypes. Immunized mice were challenged with a different GAS serotype, strain M12, and, as expected, CFUs in the M12 challenged mice were almost 10^2^-fold lower than those in adjuvant control mice (Figure 4B). The role of anti-Sda1 was also tested in a GAS systemic infection model. Sda1-immunized mice were challenged i.n. with a high dose of M1 and lethality of mice was monitored over 14 days. We found a similar lethality between Sda1-immunized and CPG control groups (Figure 4C). These results indicated that Sda1 vaccination could prevent GAS from colonization at the nasopharynx efficiently and immunity able to block additional concomitant virulent factors other than Sda1 is required for protection against systemic or invasive GAS infection.

### 3.5. Anti-Sda1 Prevented NETs Degradation and Mucosal Infection by the Sda1-Negative M3.11 Strain

GAS produce several distinct secreting DNases, including Sda1, SpdB, Sda2, Spd1, Spd3, Spd4, Sdn, and DNase B (SpeF) [9,27]. Amino acid sequence alignments analysis by ClustalW showed that all these DNases share a highly conserved domain required for DNase activity [10]. Strain M3.11 does not express Sda1 (molecular weight approximately 50 kD). The plasmid degradation assay revealed that the culture supernatant of M3.11 digested DNA in a dose-dependent manner (Appendix A), indicating that M3.11 produces molecules with DNase activity other than Sda1. Amino acid alignment analysis of the proteome encoded by M3.11 genomic DNA showed that M3.11 contains DNases identical with DNase B (25.8 kD) and SpdB (25.3 kD) (Appendix A), and DNase B shares 75% similarity with Sda1. As shown in Figure 1B, an anti-Sda1 recognized a weak band (approximately 25 kD) in all tested strains, including strain M3.11, suggesting that these bands might be other DNases and anti-Sda1 serum may prevent DNase activity of strain M3.11. As shown in Figure 5A, mouse Sda1 serum rescued the degradation of plasmid DNA mediated by the M3.11 culture supernatant. The NETs degradation inhibition assay showed that, similar to the M1 culture supernatant, the supernatant from M3.11 culture degraded NETs and the degradation was prevented by rabbit anti-Sda1 IgG (Figure 5B). To confirm Sda1 anti-serum can recognize other GAS DNases, recombinant DNase B, a DNase frequently expressed in most GAS strains [28], was generated. Western blot revealed that Sda1 antiserum cross-reacted with the recombinant DNase B (26.7 kD) (Figure 5C). To rule out a possible cross-reaction of anti-Sda1 with human DNases, analysis of whole protein and the catalytic site sequences between Sda1 and Homo sapien DNase I was performed and revealed no similarity between Sda1 and Homo sapien DNase I, indicating that DNases between vertebrates and bacteria are completely different. Western blot revealed that, consistent with the sequence analysis, Sda1 anti-serum did not cross-react with recombinant human DNase I (Figure 5D), indicating that anti-Sda1 cannot inhibit host DNases. These results indicated that anti-Sda1 antibody could neutralize DNases with a conserved functional domain in GAS but not hosts.

The effect of anti-Sda1 on prevention of M3.11-mediated NETs degradation suggested that Sda1 may play an active role in protection against M3.11 infection. Sda1-immunized mice were i.n. challenged with M3.11, and CFUs in NALT were determined 24 h after challenge. We found that, similar to the M1-challenged mice, following the Sda1- strain M3.11 challenge, CFUs from Sda1-immunized were 10^2^-fold lower than those from CpG control mice (Figure 5E). These results demonstrated that Sda1-induced immunity could provide a DNase domain-based protection against GAS infection.

## 4. Discussion

The transmission and acquisition of phage-encoded DNase is considered to be associated with the emergence and global dissemination of the highly virulent M1T1 clone [10,15]. Sda1 is a potent DNase that widely distributes among epidemic strains for escaping from host immune clearance by degrading NETs [8], which suggests that it could be a potential vaccine target. In our study, we demonstrated that i.n. immunization with Sda1 induced a robust antibody response with the ability to prevent DNase-mediated degradation of NETs formed by neutrophils in vitro and, further, protected against pharyngeal colonization by GAS independent of the GAS serotypes.

High levels of Sda1-specific serum IgG and mucosal IgA were induced following i.n. immunization with Sda1. However, the anti-Sda1 antibody response was undetectable in GAS-infected mice and was as low as that to a non-bacterial antigen in the serum samples from children that had experienced GAS infection. These observations suggested that the antibody response to Sda1 is restrained during GAS natural infection or quickly disappears. Whether the lack of antibody response to Sda1 correlates with the high risk of recurrent GAS pharyngeal colonization requires further investigation.

Anti-Sda1 prevents DNase-mediated degradation of NETs. A single GAS strain expresses one or more DNases and at least six phage-encoded secreting DNases have been identified [8,9]. Amino acid sequence analysis indicates that Sda1 shared two highly conserved catalytic domains with other five DNases expressed by GAS [10]. We found that the culture supernatant of the Sda1 strain M3.11 mediated the degradation of NETs, which was neutralized by anti-Sda1, indicating that secreting DNases other than Sda1 contribute to the degradation and anti-Sda1 can target the conserved catalytic domains of these DNases. Phage-encoded DNases with homologous regions can spread among different GAS strains and even other streptococcal species [9]. Our study suggests that Sda1 is a rational choice as a vaccine candidate to target DNase-mediated neutrophil resistance.

The pharynx and tonsils are common infection sites and reservoirs for spreading of GAS in human [29]. Therefore, reducing GAS colonization in the nasal pharynx cavity would limit the transmission of the bacteria and the subsequent pathogenesis. Our results showed that i.n. immunization with Sda1 efficiently reduced GAS colonization by heterologous GAS strains.

GAS produce distinct DNases which are important for bacterial escape from the innate immune system. SpnA is a cell-wall-anchored DNase commonly expressed by all examined GAS strains [9,30]. Studies have revealed that the overall DNase activity of the SpnA deletion mutant is not significantly decreased compared with the wild-type strain [30], indicating that other secreted DNases significantly contribute to its overall activity. The high homology shared by phage-encoded DNases and the ability of broadly neutralizing NETosis resistance by anti-Sda1 suggest that anti-Sda1 immunity could prevent neutrophil resistance caused by a set of phage-encoded DNases. Further investigations are needed to define the cross-neutralizing activity of anti-Sda1 for each of other phage-encoded secreting DNases.

Sda1-immunized mice prevented pharyngeal colonization, but this did not reduce the mortality rate of systemically infected mice. Similarly, i.n. immunization with C5a peptidase (SCPA), a GAS virulence factor involved in neutrophil resistance that prevents nasopharyngeal colonization of mice by GAS [31,32], does not protect against systemic infection as efficiently as GAS-experienced mice [18]. These observations indicate that resistance to neutrophil killing contributes significantly to nasopharyngeal colonization and additional pathogenic mechanisms other than neutrophil resistance may be required for the progression of systemic infection. On the other hand, immunization with multiple virulence factors provided efficient protection against invasive GAS infection [19,33], suggesting that the efficient protection against invasive GAS infection needs the neutralization of multiple virulence factors involved in different events of the entire invasive pathogenesis.

A mutant with inactivated chromosomal and phage-encoded DNases is cleared from a skin infection site faster than the wild-type strain [8], and mice challenged subcutaneously with an Sda1-deletion mutant developed smaller lesions than those challenged with the wild-type strain [11]. The expression of Sda1 increases GAS resistance to neutrophil killing, and over-expression of Sda1 facilitates bacterial dissemination in vivo [11,12]. These findings indicate that DNases contribute to bacterial virulence for more efficient infection. Conversely, a non-virulent GAS strain in a mouse model of subcutaneous infection remains avirulent when the strain gains Sda1 from a hypervirulent M1T1 isolate [34], whereas GAS vaccines containing molecules or epitopes of multiple virulent factors significantly reduce the size of skin lesions and mortality following subcutaneous infection by GAS [19,35,36]. Collectively, these findings support that invasive skin infection and systemic infection by GAS result from a synergistic effect by different virulence factors. Whether Sda1 together with a multicomponent vaccine is able to enhance protection efficacy in different infection models will be further determined.

## 5. Conclusions

Our study shows that the homology of phage-encoded DNases of GAS allows Sda1-induced immunity to prevent degradation of NETs generated by different phage-encoded DNases of GAS and reduces bacterial colonization in the pharynx independent of serotypes. The results suggest that Sda1 is a potential candidate for vaccines as it reduces pharyngeal colonization.

## Figures and Tables

**Figure 1 vaccines-10-00102-f001:**
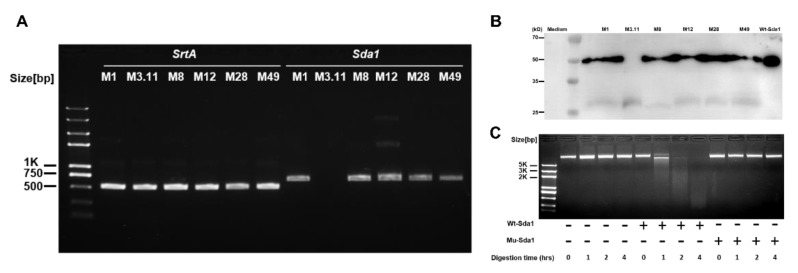
Distribution of the Sda1 genes and expression of the Sda1 protein among GAS strains. (**A**) Fragment of *Sda1* gene (684 bp) amplified by PCR analysis from six GAS strains; genomic DNA employed as a template, as indicated. The conserved *SrtA* gene (495 bp) was amplified as a control. The resulting fragments were separated by agarose gel electrophoresis. (**B**) Western blotting analysis of Sda1 expression of filtered supernatants obtained from the culture supernatant of indicated GAS strains. (**C**) Nuclease activity of recombinant Wt-Sda1 and Mu-Sda1 in degrading plasmid DNA by agarose gel electrophoresis. Nuclease activity of Wt-Sda1 resulted in a smear of degraded plasmid DNA; undegraded plasmid DNA appeared as a high molecular weight band. Data are representative of two to three independent experiments.

**Figure 2 vaccines-10-00102-f002:**
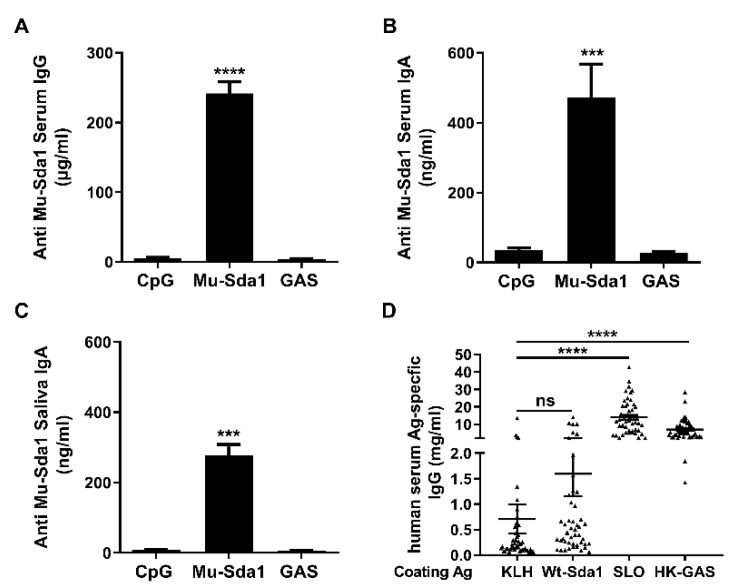
Antibody response to Sda1 in mice and humans. (**A**–**C**) Mice were immunized i.n. with Mu-Sda1 or adjuvant only or infected with GAS (Materials and Methods). Two weeks after the last immunization or infection, samples of serum and mouthwash were taken for antibody determination by ELISA. Levels of mouse serum IgG (**A**) and IgA (**B**), and IgA in mouthwash (**C**) directed to Mu-Sda1 are shown. Data are from two to three independent experiments (*n* = 9–12). (**D**) Antigen (Ag)-specific human serum IgG responses to indicated antigens (*n* = 49). Data are presented as the means ± SEM from two to three independent experiments. Statistical significance was determined by one-way ANOVA with Tukey’s post-test. *** *p* < 0.001; **** *p* < 0.0001; ns, not significant. Abbreviations: CpG, CpG-oligodeoxynucleotides; KLH, keyhole limpet hemocyanin; SLO, streptolysin O; HK-GAS, heat-killed GAS.

**Figure 3 vaccines-10-00102-f003:**
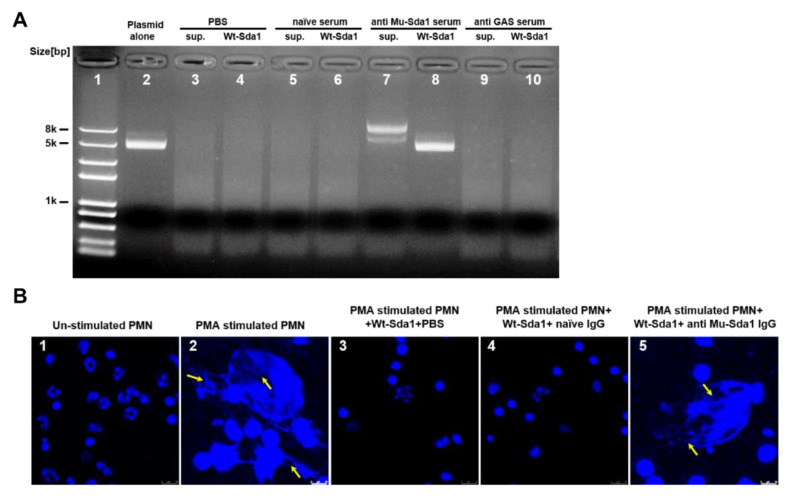
Anti-Sda1 serum prevented NETs degradation by Sda1. (**A**) Mouse anti-Sda1 serum neutralized GAS M1 culture supernatant and wild-type (Wt)-Sda1-mediated degradation of plasmid DNA. GAS M1 culture supernatant and Wt-Sda1 in the presence of an immunized serum in the plasmid degradation assay. Anti-Sda1 serum or naïve serum was pre-mixed with GAS M1 culture supernatant or Wt-Sda1 in the plasmid degradation assay. (**B**) Anti-Sda1 serum IgG neutralized Wt-Sda1-mediated degradation of neutrophil NETs. Mouse neutrophils were isolated and incubated for 4 h without stimulation (lane 1) or with PMA stimulation (lanes 2, 3, 4, 5). Released neutrophil NETs after PMA stimulation were visualized by DAPI (blue) staining. Aliquots of Wt-Sda1 were pre-incubated with the indicated rabbit serum IgG before their addition to PMA-stimulated neutrophils. Representative images from two to three experiments are shown. Scale bar, 10 mm. Abbreviations: sup., GAS culture supernatants.

**Figure 4 vaccines-10-00102-f004:**
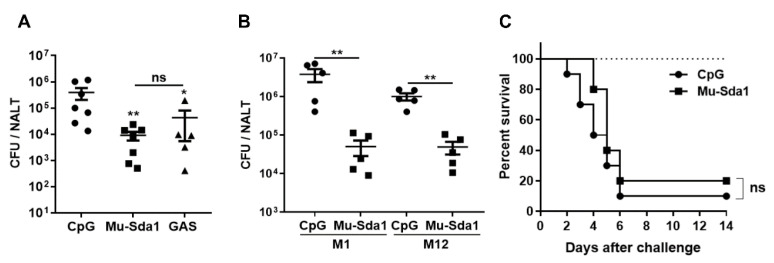
Immunization with Mu-Sda1 prevented GAS colonization in the nasopharynx. Mice were immunized as described in Figure 2. Two weeks after the last immunization, mice were challenged i.n. with GAS M1 (**A**). CFUs in NALT were determined 24 h after the challenge. (**B**) Immunized mice were challenged with GAS M1 or GAS M12, respectively. CFUs in NALT were determined 24 h after the challenge. (**C**) Two weeks after the last immunization, mice were i.n. challenged with a lethal dose of GAS M1 (3 × 10^8^). The mortality of mice was recorded for 14 days (*n* = 20–21). Data are from two independent experiments. Statistical significance was determined by one-way ANOVA with Tukey’s post-test (**A**) or two-tailed unpaired Mann–Whitney *U* nonparametric *t*-test (**B**) or log-rank test (**C**). * *p* < 0.05; ** *p* < 0.01; ns, not significant.

**Figure 5 vaccines-10-00102-f005:**
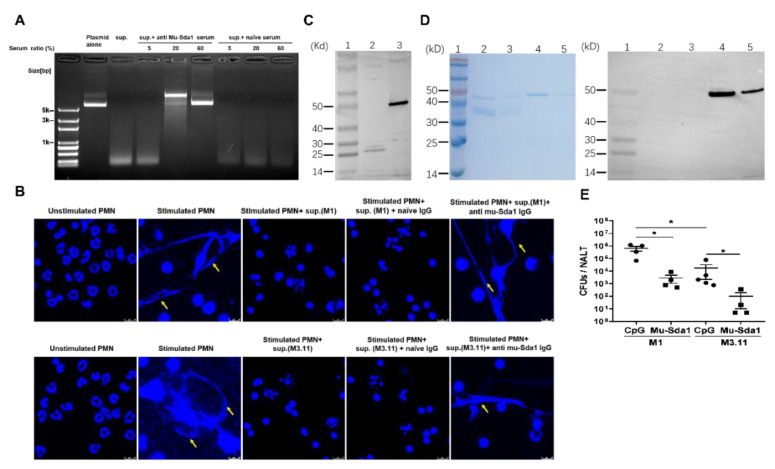
Anti-Sda1 prevented NETs degradation and mucosal infection by Sda1-negative M3.11 strain. (**A**) Sda1 immunized serum neutralized nuclease activity in GAS M3.11 culture supernatants in the plasmid degradation assay. (**B**) Anti-Sda1 serum cross-neutralized DNases expressed in the GAS M1 and M3.11 culture supernatant. Isolated mouse neutrophils were stimulated and stained with DAPI as described in Figure 3. Diluted GAS M1 and M3.11 culture supernatant were pre-incubated with rabbit immunized serum IgG before incubation with neutrophil NETs. Representative images are shown. Scale bar, 10 μm. (**A**,**B**) Experiments were performed in triplicate. (**C**) Recombinant DNase B was analyzed by SDS-PAGE and Western blotting with anti-Sda1 serum. Lane 1, Mw; lane 2, recombinant DNase B; lane 3, recombinant Sda1. (**D**) Recombinant Homo sapien DNase I was analyzed by SDS-PAGE and Western blotting with anti-Sda1 serum. Left panel: SDS-PAGE, right panel: Western blot. Lane 1, Mw; lane 2, recombinant DNase I (300 ng); lane 3, recombinant DNase I (100 ng); lane 4, recombinant Sda1 (300 ng); lane 5, Recombinant Sda1 (100 ng). (**E**) Mice were immunized as described in Figure 2. Two weeks after the last immunization, mice were i.n. challenged with GAS M1 or M3.11 strain, respectively. CFU in NALT were determined 24 h after challenge. (*n* = 5). Statistical significance was determined by two-tailed unpaired Mann–Whitney *U* nonparametric *t*-test (**D**). * *p* < 0.05. Abbreviations: Mw, molecular weight.

## Data Availability

All data that this study is based upon are available from the corresponding author upon request.

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
