# Peer review of "Immunity to Sda1 Protects against Infection by Sda1+ and Sda1 Serotypes of Group A Streptococcus"

_vaccines, 2022, doi:10.3390/vaccines10010102_

Round 1

Reviewer 1 Report

This manuscript explores the potential of using an enzymatic-inactive mutant of the secreted DNase Sda1 as a vaccine target. Authors show antibody production after intranasal immunisation with recombinantly produced protein adjuvanted with CpG. Functional neutralisation of enzymatic activity with Sda1 antiserum is tested in vitro, and protective efficacy after immunisation is observed in a “colonisation”, but not invasive disease mouse model.

Overall, I found the manuscript well written and presented. It was good to see the use of multiple GAS strains in the challenge studies. However, a more complete discussion would significantly improve the quality considering the conclusion that protective efficacy is only seen against colonisation.

I have some comments that I believe, if addressed, could significantly improve the completeness of the study:

  1. Regarding the sub-lethal dose used to model colonisation: It is not clear if colonisation is achieved since mice are sacrificed after 24h in this model.
  2. Expression of Sda1 was shown by western blot of a number of invasive disease isolates. What is the expression pattern like in pharyngeal isolates? How well conserved is the gene globally?
  3. Could authors please explain the difference between Fig 1b and 5A? Is this the same image, just cropped/different exposure?
  4. 2D – is the serum all from invasive disease patients? Is there any distinction from the 10 or so patients with relatively high Sda1 antibodies?
  5. Fig 3A – why is the band in lane 7 higher than control and lane 8? Similar question for fig. 5B
  6. Line 293 – some of these DNases are not described in the referenced article, at least not in the nomenclature used. For clarity, please provide the new nomenclature.
  7. 5A – this very weak band is not overly convincing. Can a control for the alternative DNase be included? Or perhaps a pull-down with Sda1 antibodies coupled with mass-spec to identify the 25kDa band?
  8. Is the lack of protective efficacy against invasive disease due to the immunisation route? Please discuss.
  9. Could authors please discuss the possible implications between their results and other published studies on Sda1 i.e. Walker et.al. has shown that Sda1 is upregulated in invasive disease.

Author Response

Reviewer 1

Comments and Suggestions for Authors

This manuscript explores the potential of using an enzymatic-inactive mutant of the secreted DNase Sda1 as a vaccine target. Authors show antibody production after intranasal immunisation with recombinantly produced protein adjuvanted with CpG. Functional neutralisation of enzymatic activity with Sda1 antiserum is tested in vitro, and protective efficacy after immunisation is observed in a “colonisation”, but not invasive disease mouse model.

Overall, I found the manuscript well written and presented. It was good to see the use of multiple GAS strains in the challenge studies. However, a more complete discussion would significantly improve the quality considering the conclusion that protective efficacy is only seen against colonisation.

I have some comments that I believe, if addressed, could significantly improve the completeness of the study:

  1. Regarding the sub-lethal dose used to model colonisation: It is not clear if colonisation is achieved since mice are sacrificed after 24h in this model.

GAS is a human pathogen. Mice are not sensitive to GAS infection unless a high dose is used. This model has been used for studying GAS pathogenesis and immune response to the bacteria because it mimics natural infection route of GAS. Previous studies have shown that at the dose we used, inoculated bacteria can stay in the nasopharynx (NALT) for several days with the highest number at 24 h and gradually be cleared [1]. Also immunized mice show significant CFU reduction in NALT after challenge[2-4]

  1. Expression of Sda1 was shown by western blot of a number of invasive disease isolates. What is the expression pattern like in pharyngeal isolates? How well conserved is the gene globally?

We appropriate the reviewer for the question. The Sda1 gene is found in 135 of 141 emm12 clinical isolates, which are the most frequently isolated serotype from cases with scarlet fever [5]. The blast search result shows that the identity of the Sda1 amino acid sequence is >77.64% among examined GAS strains. The expression pattern of Sda1 in pharyngeal isolates and the association of Sda1 with pharyngeal colonization are not known.

  1. Could authors please explain the difference between Fig 1b and 5A? Is this the same image, just cropped/different exposure?

They are different images from two independent experiments. However, they give the same information. We thank the reviewer’s question very much. Fig. 1B is replaced by Fig. 5A and Fig. 5A is removed. The required information in the text refers to the new Fig. 1B (page 9, lines 315-318).

  1. 2D – is the serum all from invasive disease patients? Is there any distinction from the 10 or so patients with relatively high Sda1 antibodies?

The serum samples are not all from patients with invasive disease. SLO is GAS specific. High levels of SLO antibodies represent an experienced GAS infection. However, the same 49 patients with relatively higher levels of SLO antibody showed Sda1 antibody as low as a non-bacterial antigen (KLH). These results suggest that compared to SLO and other GAS surface molecules (HK-GAS), human response to Sda1 is relatively weaker or doesn’t stay a long time. As the reviewer noticed that some of the patients showed higher Sda1 antibody levels; however, they still fall in the range of antibody response to KLH. Therefore, it likely reflects individual differences. We see the reviewer’s point that Sda1 might be related to pharyngeal infections since Sda1 immunization reduced GAS colonization in NALT. More defined clinical samples are required to answer the question.

  1. Fig 3A – why is the band in lane 7 higher than control and lane 8? Similar question for fig. 5B

            The possible reason could be that the anti-Sda1 antibody did not completely neutralize other GAS DNases with less homology with Sda1. The non-neutralized DNase cleaves the circular plasmid into linearized and migrated slower. The higher bands could be partially cleaved plasmid DNA, which migrated slower.

  1. Line 293 – some of these DNases are not described in the referenced article, at least not in the nomenclature used. For clarity, please provide the new nomenclature.

            The nomenclature of DNases is changed according to reference 9 in the manuscript and related text is modified accordingly (page 8, lines 304-305; page 9, lines 314-315; page 12, line 422; Supplementary Figure 3).

  1. 5A – this very weak band is not overly convincing. Can a control for the alternative DNase be included? Or perhaps a pull-down with Sda1 antibodies coupled with mass-spec to identify the 25kDa band?

            Fig. 5A now is Fig. 1B in the revised manuscript. We major focus of doing this experiment is to answer if anti-Sda1 could recognize DNases expressed in GAS culture supernatant and be responsible for the neutralizing efficacy of the antiserum. The weak bands suggest that they might be other GAS DNases that share lower homology with Sda1. The suggested pull-down assay is a good approach to try. We thank the reviewer for this suggestion and would like to do it in our future studies.

  1. Is the lack of protective efficacy against invasive disease due to the immunisation route? Please discuss.

We previously reported that a single conserved molecule sortase A (SrtA) or combined SrtA with SCPA (streptococcal C5a peptidase) induces serotype-independent protection against GAS in the URT mucosa in the same mouse model used in this study[2,3]. However, SrtA/SCPA immunization did not protect against invasive infection until three more virulence factors are added to the vaccine (5CP) [4]. These results suggest that the efficient protection against invasive infection needs the neutralization of multiple virulence factors involved in different events of the entire invasive pathogenesis. Also, parenteral immunization with 5CP failed to reduce mucosal GAS colonization, despite high levels of serum IgG [4]. Because human tonsils act as a reservoir and entry port for GAS, reducing GAS colonization at mucosal sites by secretory IgA would limit the chances of dissemination and contribute to protection against systemic infection and lethality.

  1. Could authors please discuss the possible implications between their results and other published studies on Sda1 i.e. Walker et.al. has shown that Sda1 is upregulated in invasive disease.

We appreciate the reviewer’s suggestion, and add related content in the Discussion section of the revised manuscript (page 12, lines 403-404).

References

  1. Park, H.S.; Francis, K.P.; Yu, J.; Cleary, P.P. Membranous cells in nasal-associated lymphoid tissue: a portal of entry for the respiratory mucosal pathogen group A streptococcus. J Immunol 2003, 171, 2532-2537, doi:10.4049/jimmunol.171.5.2532.
  2. Chen, X.; Li, N.; Bi, S.; Wang, X.; Wang, B. Co-Activation of Th17 and Antibody Responses Provides Efficient Protection against Mucosal Infection by Group A Streptococcus. PLoS One 2016, 11, e0168861, doi:10.1371/journal.pone.0168861.
  3. Fan, X.; Wang, X.; Li, N.; Cui, H.; Hou, B.; Gao, B.; Cleary, P.P.; Wang, B. Sortase A induces Th17-mediated and antibody-independent immunity to heterologous serotypes of group A streptococci. PLoS One 2014, 9, e107638, doi:10.1371/journal.pone.0107638.
  4. Bi, S.; Xu, M.; Zhou, Y.; Xing, X.; Shen, A.; Wang, B. A Multicomponent Vaccine Provides Immunity against Local and Systemic Infections by Group A Streptococcus across Serotypes. mBio 2019, 10, doi:10.1128/mBio.02600-19.
  5. Davies, M.R.; Holden, M.T.; Coupland, P.; Chen, J.H.; Venturini, C.; Barnett, T.C.; Zakour, N.L.; Tse, H.; Dougan, G.; Yuen, K.Y.; et al. Emergence of scarlet fever Streptococcus pyogenes emm12 clones in Hong Kong is associated with toxin acquisition and multidrug resistance. Nat Genet 2015, 47, 84-87, doi:10.1038/ng.3147.

Reviewer 2 Report

This article describes the analysis of the group A streptococcus DNase Sda1 as a potential vaccine candidate. The authors show that intranasal immunisation with recombinant non-functional Sda1, but not GAS infection, results in specific mucosal antibody responses (both IgG and IgA). They further show that rSda1 destroys neutrophil extracellular traps in vitro and this can be inhibited with serum from rSda1-immunised animals. Lower CFUs were detected in the NALT of vaccinated mice in a mouse pharyngitis model. Finally, the authors argue that the antibodies raised against Sda1 cross-protect sda1-negative strains by neutralising other DNases.

The manuscript is generally well written and the methodology is adequate (apart from last section). The results are novel and of interest.

Specific points:

  • Line 106: why was mouse serum used, not rabbit serum? It is sometimes confusing in the result section what serum was used.
  • Do the mice actually develop pharyngitis? Would the model be better described as ‘colonisation model’?
  • GAS is generally not a very good coloniser of the mouse nasopharynx. In this context, it would be important to describe how the bacteria are recovered. The method section states “as previously described” and the cited literature says the same citing another article that again includes that statement.
  • Have the authors experimentally determined the CFUs used for the mouse challenge? There is a small window between the non-lethal challenge (2*10E8 CFU) and the lethal challenge (3*10E8 CFU).
  • How frequently is the sda1 gene found in GAS strains and how much is it conserved? The authors argue that it is commonly found (line 196) but only 6 strains were analysed. This could be done by a BLAST search.
  • Human serum was analysed from 49 children. What was the health status of the children? Anti-SLO titers (used as a control) are known to last for quite long (a limitation of the ASLO serology).
  • Line 225: the authors argue that non detectable Sda1 antibody in GAS-infected mice and the limited anti-Sda1 response in humans indicate the humoral response to Sda1 is restrained in natural GAS infection. Isn’t it possible that Sda1 is not expressed under the experimental conditions (mice) and that the children were not recently infected.
  • Line 263: reference 25 doesn’t provide strong evidence that NETs play a major role in pharyngeal infection. The authors of this article acknowledged their study had limitations such as; NETs were not directly detected in vivo in this model and the model required large number of GAS and was lethal (unlike pharyngitis).
  • Section 3.5: There is no way of telling that the 25kd band in the Western-blot represents another DNAse. What are the aa similarities between Sda1 and other secreted DNases? The authors should have generated recombinant proteins of other DNases (they have the experience and technology) and test for cross-reactivity and cross-neutralisation.
  • Discussion: the authors should emphasise that NET killing was demonstrated in vitro and not in vivo.
  • Line 350: if the ab raised against rSda1 neutralising DNase activity by binding to the catalytic site, is there a possibility that the ab would also inhibit host DNases as the catalytic site of DNases is well conserved?
  • The discussion section lacks flow and is sometimes a bit vague, eg. “Immunity against a set of virulence factors involved in the infection progression significantly reduces the mortality of systemically infected mice”.
  • Figures legends: please add abbreviations, e.g. figure 2: SLO, HK-GAS, KLH, CpG

Reviewer 3 Report

The current manuscript presented by Bi et al. describes an interesting study on immunity to the DNase Sda1 and protection against nasal and systemic streptococcus infection. This is a very well put together and well written manuscript with convincing data and I have only some minor comments:

  • It is not clear to me if and when the rabbit sera has been used, if should be indicated more clearly if it was used. I assume it was used when the authors refer to anti-Sda1 IgG, but since this could also refer to sera from the immunized mice I'm not sure.
  • Similarly, in line 151-157, the authors state tha "a rabbit" was bought, but multiple "rabbits" were immunized, please specify. Please also indicate what the control serum was used for, if it applies, or remove from the text.
  • The axis range in Figure 2 should be the same in A-C to make comparison easier.
  • There are a few typos, e.g. missing spaces: Sda1induced (line 67), Sda1by (line 237), Wt-Sda1was (line 242), M3.11culture (line 306); line 66 should be "intranasal"; line 218 should read "that were infected with GAS"
  • The label "Mw" should be removed from the figures when DNA ladders are shown, the label "Size [bp]" is sufficient and more correct.

Author Response

Reviewer 3

Comments and Suggestions for Authors

The current manuscript presented by Bi et al. describes an interesting study on immunity to the DNase Sda1 and protection against nasal and systemic streptococcus infection. This is a very well put together and well written manuscript with convincing data and I have only some minor comments:

  1. It is not clear to me if and when the rabbit sera has been used, if should be indicated more clearly if it was used. I assume it was used when the authors refer to anti-Sda1 IgG, but since this could also refer to sera from the immunized mice I'm not sure.

We are sorry about this confusion. Both mouse and rabbit antisera were used in the study (anti-mu-Sda1). Considering the quantity of antiserum required for experiments rabbit antiserum and also anti-Sda1 rabbit IgG were prepared for a shortage of mouse antiserum. In our study, generally, rabbit IgG was used in NETs degradation assay (page 4, lines 176-177; page 7, lines 259-260; page 8, line270; page 9, line 324), and mouse antiserum in plasmid DNA degradation assay (page 7, line 254; page 9, line 320). We specify that in the related figure legends.

  1. Similarly, in line 151-157, the authors state tha "a rabbit" was bought, but multiple "rabbits" were immunized, please specify. Please also indicate what the control serum was used for, if it applies, or remove from the text.

Thanks for pointing out the mistake. It is corrected (page 4, lines 153-154 and line 159).

  1. The axis range in Figure 2 should be the same in A-C to make comparison easier.

The axis of Figures 2B and C are rescaled for easier comparison of IgA in serum and saliva samples. We remain the axis ranges of Figure 2A and D since they are IgG and from different species (mouse and human).

  1. There are a few typos, e.g. missing spaces: Sda1induced (line 67), Sda1by (line 237), Wt-Sda1was (line 242), M3.11culture (line 306); line 66 should be "intranasal"; line 218 should read "that were infected with GAS"

Thanks for pointing out the mistakes. These are corrected in the revised manuscript (page 2, line 67; page 7, line 248; page 7, line 253; page 10, line 323; page 2, line 66; page 6, line 228).

  1. The label "Mw" should be removed from the figures when DNA ladders are shown, the label "Size [bp]" is sufficient and more correct.

We thank the reviewer again. "Mw" is removed from the figures in the revised manuscript (page 5, Figure 1A - C; page 7, Figure 3A; page 10, Figure 5A; Supplementary material, Figure S2).

Round 2

Reviewer 2 Report

This is an improved version of the original manuscript and the author have addressed some of my concerns. However, a major issue remains. The authors added the information “The Sda1 gene was reported in 135 isolates of 141 emm12 GAS clinical strains from Hong Kong clinical isolates [2]. The blast search result shows that the identity of Sda1 amino acid sequence is >77.64% with other strains of Streptococcus pyogenes.” All of the Hong Kong isolates are emm12 type. Sda1 is not found in many important GAS strains including serotype M3, M5, M28, M49 and M89 and is often poorly conserved in other strains. A Sda1 vaccine would therefore have only very limited strain coverage. In this context the hypothesis that a Sda1 vaccine could cross-protect due to neutralisation of other secreted DNase becomes critical, but unfortunately the manuscript doesn’t provide any conclusive evidence for that. At least, the authors need to show that there is cross-reactivity of anti-Sda1 ab with other proteins (even though this wouldn’t necessarily mean cross-protection).

Required revisions:

  1. Generate recombinant versions of other secreted DNases (I believe, DNaseB is also commercially available) and test for cross-reactivity with anti-Sda1 ab.

Minor point:

The authors state that “No significant similarity is found between Sda1 and Homo sapien protein sequences by blast analysis (https://blast.ncbi.nlm.nih.gov/Blast.cgi), suggesting that anti-rSda1 cannot inhibit host DNases”. However, the catalytic site of DNAses is usually conserved, so if the anti-Sda1 ab neutralises the enzymatic activity, it might recognise the catalytic site of e.g. DNaseI.

Author Response

Response to reviewer 2 (Round 2)

This is an improved version of the original manuscript and the author have addressed some of my concerns. However, a major issue remains. The authors added the information “The Sda1 gene was reported in 135 isolates of 141 emm12 GAS clinical strains from Hong Kong clinical isolates [2]. The blast search result shows that the identity of Sda1 amino acid sequence is >77.64% with other strains of Streptococcus pyogenes.” All of the Hong Kong isolates are emm12 type. Sda1 is not found in many important GAS strains including serotype M3, M5, M28, M49 and M89 and is often poorly conserved in other strains. A Sda1 vaccine would therefore have only very limited strain coverage. In this context the hypothesis that a Sda1 vaccine could cross-protect due to neutralisation of other secreted DNase becomes critical, but unfortunately the manuscript doesn’t provide any conclusive evidence for that. At least, the authors need to show that there is cross-reactivity of anti-Sda1 ab with other proteins (even though this wouldn’t necessarily mean cross-protection).

Required revisions:

  1. Generate recombinant versions of other secreted DNases (I believe, DNaseB is also commercially available) and test for cross-reactivity with anti-Sda1 ab.

We appreciate the reviewer’s point. The recombinant DNase B was generated as suggested. Western blot showed that Sda1 antiserum reacted with the recombinant DNase B (26.7Kd) as shown in Figure 5C. The results are added in the revised manuscript (page 10, lines 346-349; Figure 5C and Figure legend, lines 372 to 376).

Minor point:

The authors state that “No significant similarity is found between Sda1 and Homo sapien protein sequences by blast analysis (https://blast.ncbi.nlm.nih.gov/Blast.cgi), suggesting that anti-rSda1 cannot inhibit host DNases”. However, the catalytic site of DNAses is usually conserved, so if the anti-Sda1 ab neutralises the enzymatic activity, it might recognise the catalytic site of e.g. DNaseI.

We previously found no significant similarity in the whole protein sequences between Sda1 and Homo sapien DNase I by blast analysis. Further amino acid sequence alignments analysis of the catalytic site by ClustalW showed no similarity between Sda1 and Homo sapien or mouse Dnase I, indicating that DNases between vertebrates and bacteria are completely different. To be more confirmed, Western blot was performed and revealed that consistent with the sequence analysis, Sda1 antiserum did not cross-react with recombinant human DNase I (Figure 5D), indicating that anti-Sda1 cannot inhibit host DNases. The results are added in the revised manuscript (page 10, lines 349-354, Figure 5D).

Round 3

Reviewer 2 Report

The authors have addressed all my points. 

One very minor issue:

Line 324: "Human expresses DNases". A somehow weird sentence. Please rephrase.

Author Response

Response to Reviewer 2 Comments (Round 3)

Comments and Suggestions for Authors

The authors have addressed all my points.

One very minor issue:

Line 324: "Human expresses DNases". A somehow weird sentence. Please rephrase.

Response: We thank the reviewer’s suggestion. This sentence is removed and rewritten in the revised manuscript (page 8, lines 328-331).